# Blockchain Traceability Adoption in Low-Carbon Supply Chains: An Evolutionary Game Analysis

**Chen Zhang [1], Yaoqun Xu [2,]*** and **Yi Zheng [3]**

1    School of Management, Harbin University of Commerce, Harbin 150028, China; zchh2022@163.com
2    Institute of Systems Engineering, Harbin University of Commerce, Harbin 150028, China
3    School of Computer and Information Engineering, Harbin University of Commerce, Harbin 150028, China; zhengy@s.hrbcu.edu.cn
*    Correspondence: xuyq@hrbcu.edu.cn

**Abstract:** Blockchain technology has brought innovation to supply chain management, particularly in managing carbon emissions in the manufacturing sector. However, there is a research gap regarding the policy tools and the role of local governments in implementing blockchain technology to achieve carbon emissions traceability. Additionally, the strategic relationships and policy implications resulting from the implementation of blockchain technology are not examined systematically. An effective method for examining the strategies used in interactions between supply chain stakeholders and governments is evolutionary game theory, or EGT. This paper employs mathematical modelling and MATLAB 2016 software simulation to examine the decision-making process of manufacturing companies when considering implementing blockchain technology traceability. Specifically, the subjects in the model include product manufacturers (PM), product suppliers (PS), and local governments (LGs). The aim is to examine the decision-making behavior of carbon traceability participants in blockchain technology. This paper analyses the most effective blockchain-based traceability strategies for low-carbon supply chain members under a variety of scenarios by modifying the parameters. The findings suggest the following: (1) Manufacturers and suppliers need to manage the cost of blockchain traceability, collaborate to create an environmentally friendly product certification system, and improve brand image. (2) Local governments should set up efficient reward and punishment systems to incentivize supply chain stakeholders to engage in the blockchain traceability system. The aforementioned discoveries furnish policymakers with guidance to encourage the implementation of blockchain-based carbon footprint traceability technology, thereby establishing a transparent carbon footprint traceability framework across the entire supply chain.

**Keywords:** low-carbon chain; blockchain; carbon footprint; simulation analysis

## 1. Introduction

In recent decades, the global greenhouse effect has intensified, becoming a major concern for the international community. Key initiatives to support the sustainability of the global economy are energy efficiency, emissions reduction, and the shift to a low-carbon economy [1–3]. Manufacturing enterprises and their supply chains are considered to be the most significant contributors to carbon emissions, and low-carbon supply chains have become the new focus of global manufacturing [4,5]. By using the Internet of Things (IOT) to track carbon emissions during product manufacturing, transportation, and distribution, blockchain technology is able to provide traceability of emissions more effectively than traditional automation methods utilizing QR and RFID codes [6–8]. Therefore, blockchain with distributed ledger technology, immutable records, and smart contracts can provide a solution for CF traceability in supply chains, and CF traceability based on blockchain is of great significance to building sustainable supply chains and promoting carbon emission reductions for focus enterprises [9–11].

By offering traceability services for tea and cold chain goods, the blockchain traceability platform (Baas), developed by ANT GROUP, has made food supply chain traceability a reality [12]. Blockchain technology offers a unique answer to the low-carbon supply chain management problem, notwithstanding its diverse applications in traceability. All supply chain participants must actively participate in the complicated applications of blockchain technology [13,14]. Concerns from certain stakeholders could include the price of personnel training and software integration. It is noteworthy that, with regard to supply chain stakeholders' decision-making behavior, there is still a dearth of research on the tactics used by relevant companies to deploy blockchain technology, despite the significance of blockchain technology in creating carbon traceability solutions [13].

Through review of the literature and analysis, the focus of the existing study is on the characteristics and contributions of blockchain technology, carbon performance of sustainable economies, and the advantages and challenges of blockchain traceability, and it pays little attention to the gaming behavior of participants when deploying blockchain-technology-based carbon footprint traceability in the supply chain [9,11,15]. Therefore, there are several gaps in the research: Firstly, the existing literature on supply chain traceability often overlooks the impact of government policy regulation and supervision on the process, instead focusing primarily on the participants involved. Secondly, to analyze behavioral decisions in technology adoption, some scholars have used evolutionary game theory; free-riding behavior is rarely considered. The prevention of supply chain participants' opportunistic conduct in blockchain traceability has emerged as a critical concern.

On the one hand, neither the PMs nor the PSs are totally rational; each decision can only be made on the basis of the limited information available to them. It is also disadvantageous for supply chain member companies to invest in traceability systems time and again due to the high cost associated with traditional data migration and the potential lack of adoption of new technologies. On the other hand, by participating in the construction of the blockchain carbon emission traceability system, each member could obtain corresponding benefits. The PMs can reduce carbon management costs by identifying and improving production links that consume large amounts of energy, while promoting a low-carbon brand image to draw in more customers and split the blockchain technology's cost [16]. The PSs can access the blockchain traceability platform to obtain product lifecycle carbon emission information and provide traceability platform certification services. Through a strict regulatory reward and punishment mechanism, the LGs motivate supply chain players to aggressively embrace blockchain traceability, rationally design carbon allowances and taxes, and further advance the growth of the low-carbon economy [17].

Blockchain technology has made significant strides in the field of manufacturing. However, low-carbon manufacturing remains a challenge due to the high cost of managing carbon emissions and incomplete information, including carbon footprints, between companies up and down the supply chain [18]. In order to evaluate the dynamic process of traceable decision making among the PMs, PSs, and LGs, this paper employs a mathematical modeling technique. Specifically, under the supervision of the local government, some manufacturers produce carbon-emitting products (e.g., automotive components and construction materials) and provide them to common suppliers for circulation to consumers. In order to realize a low-carbon supply chain and improve the carbon trading market, the government needs to promote blockchain technology adoption by supply chain member companies to trace CF, and the companies will participate in the process of deploying blockchain technology from the perspective of their own interests. Consequently, a blockchain-based traceability system can document each stage of operations from the manufacturing end to the sales end and calculate their CFs based on products' lifecycle assessment (LCA).

As mentioned above, implementing a carbon traceability system for products from production to consumption requires the involvement of the PM, the PS, and the LG agencies. However, the interests of all parties are affected by the dynamics of a number of factors, which can lead to low levels of participation. EGT argues that finite rational agents reach equilibrium through continuous strategic interaction. As a result, the study of

environmental investment strategies has made extensive use of EGT [19]. Therefore, we contend that EGT is appropriate for investigating the decision-making interactions of players in supply chains with blockchain-based carbon traceability, assisting participants in making sane decisions and advancing the interests of all subjects [20]. Therefore, an analytical model of "Tripartite evolutionary game model of supply chain carbon emission traceability based on blockchain technology" is established, as shown in Figure 1. In this paper, in view of the shortcomings of the above research, the government is regarded as the main player of the game, and the influence of the government's measures for rewards and penalties on the deployment of blockchain technology is analyzed. In addition, through mathematical model analysis and simulation, we examine the "free-riding" benefits and the impact of other influencing factors such as adoption costs on the evolutionary path to arrive at policy recommendations for accelerating enterprise adoption of blockchain and avoiding "free-riding" behavior.

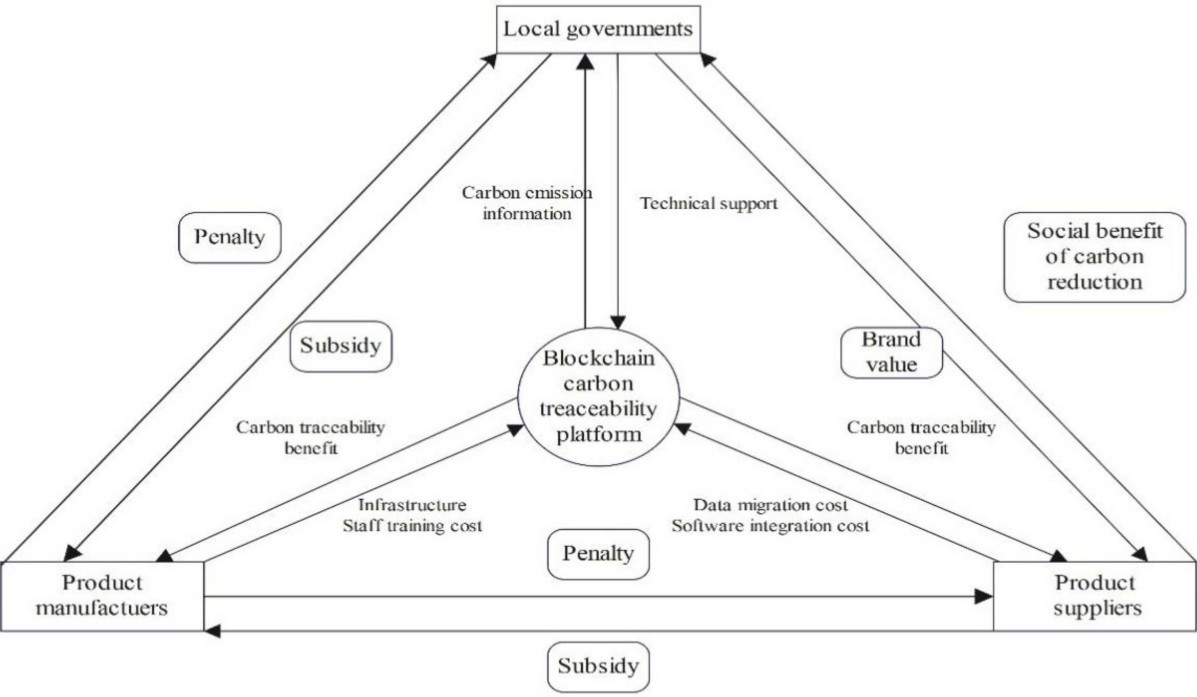

**Figure 1.** Blockchain-based tripartite supply chain evolutionary game model of carbon emission tracking.

This study's primary goals are as follows: (a) Examine the long-term decision-making behavior of the PMs, PSs, and LGs—key players in manufacturing blockchain carbon footprint traceability; (b) Investigate the primary factors influencing the creation of the manufacturing supply chain's blockchain-based carbon footprint traceability system; (c) Derive evolutionarily stable strategies (ESS) for various decision scenarios.

This paper presents the following contributions: (1) We categorize the use cases of blockchain traceability in various domains, with a focus on how blockchain technology can enable traceability of carbon emissions in supply chains. (2) We apply EGT to study the dynamic strategic decisions of the PMs, PSs, and LGs in blockchain traceability. The ideal evolutionary outcome and development path are derived. (3) It has been observed that the effectiveness of rewards and punishments can vary among different subjects. (4) The necessity of implementing blockchain technology and the brand advantage of creating a low-carbon product certification are also determined.

The remainder of the paper is as follows: Research on blockchain, carbon emission reduction, and evolutionary gaming is reviewed in Section 2. For blockchain-based deployment, Section 3 presents the tripartite EGT model of the PMs, PSs, and LGs and investigates the ESS by examining the equilibrium point's asymptotic stability. Numerical simulations

are used in Section 4 to show how the ESS behaves in various settings and how changing parameters affect these tactics. In conclusion, Section 5 makes policy recommendations.

## 2. Review of the Literature and Theoretical Structure

### 2.1. Blockchain Mechanisms

Blockchain, the decentralized distributed ledger [21], has emerged as a significant digital technology in the Industry 4.0 era [7]. The core technical features of blockchain include the following: (1) Decentralization [22]; (2) Disintermediation [23]; (3) Immutability [24]]; (4) Anonymity [25]; (5) Smart contracts [26]; (6) Traceability [27]; (7) Cost reduction; (8) Transparency [28]; (9) Security and privacy.

Traceability, a fundamental technical feature of blockchain technology, enables better transparency in supply chain management [29,30]. Its constituent conditions include decentralization, in which the blockchain adopts the peer-to-peer network model, supplying services and resources to every node, allowing every node to access a copy of the distributed ledger, and enabling any node to access transaction information [31]. The consensus mechanism ensures that data are put into the blockchain and, after gaining the consent of the majority of nodes through the consensus algorithm, guarantees the consistency of each transaction among all nodes [18]. The immutable property refers to the fact that a single block in the blockchain includes the activities from the previous block as well as a timestamp, i.e., "harsh" [32]; so, once the block is formed, the information cannot be changed. At the same time, it is used in conjunction with other digital technologies to collect product-related information, namely RFID and IoT [21]. In brief, traceability facilitates transparency in the supply chain through the monitoring and documentation of movements. Table 1 presents an overview of scholars' research on the application of traceability across distinct fields.

**Table 1.** Summary of the many fields' traceability research.

| Authors | Fields | Highlights | Effect |
|---|---|---|---|
| (Ahmed and MacCarthy, 2021) [33] | Textile and Apparel | Clarified the objectives of the traceability program and properly defined the scope of the traceability solution. | Enriched the discussion on key supply chain traceability considerations and the scope of product identification throughout the supply chain. |
| (Casino and Kanakaris, 2021). [34] | Dairy Sector | Developed and tested a distributed, trusted, and secure architecture for the FSC traceability system. | Demonstrated the applicability and overall benefits of the model through the development of fully functional smart contracts and a local private blockchain. |
| (Wang and Wang, 2020) [35] | Precast Construction | Established a novel blockchain traceability information management framework. | Solved the problems of automated information sharing, traceability, and visibility in a precast supply chain. |
| (Zheng and Xu, 2023) [36] | Farm Commodities | Examined the decision-making process for blockchain adoption traceability in agriculture. | Analyzed the key factors for implementing a blockchain-enabled agricultural product traceability system and made policy recommendations. |

### 2.2. Carbon Footprint Management in Supply Chain

As a system definition, the overall amount of carbon dioxide emissions generated by an activity, whether direct or indirect, or the total amount of emissions accumulated over the lifecycle of products is measured as the carbon footprint. To perform a carbon footprint analysis using a lifecycle assessment (LCA), organizations throughout the supply chain must estimate a product's complete lifecycle, including its direct and indirect carbon dioxide emissions [37]. Many researchers utilize LCA to quantify the carbon footprint of the whole supply chain.

A prevailing strategy in CF management involves transitioning from direct impact reporting on field processes to indirect impact reporting on a company's upstream supply chain or downstream product use and handling. Analysis results demonstrate that supply chain Scope 3 indirect emissions contribute to 56.5% of all carbon emissions across industries, emphasizing the significance of including carbon footprint management throughout

the entire supply chain [38]. Scholars have integrated carbon footprints into decision-making models for purchasing, production, and inventory management. They have discovered that cooperation between supply chain members can effectively reduce carbon emissions [39]. By conducting an empirical investigation on HMCS's front bumper products and their key PSs, the focus company may ascertain the overall carbon footprint value of individual vehicle units by determining and quantifying the carbon footprint of important parts and goods from important vendors [38]. However, there is a gap in measuring carbon emissions in Scope 3 in the manufacturing sector, endangering integrity. No suitable solution has been found for the allocation of carbon footprint costs [40].

*2.3. Blockchain-Based Carbon Emission Traceability in Supply Chain*

The capacity to track and retrieve information and identify commodity records is known as traceability in the supply chain through information storage systems in both direction [30]. Traceability of direct and indirect carbon emissions can help companies describe the carbon footprint of their products, thereby identifying carbon risks throughout the supply chain and improving carbon performance [38]. The key to reducing carbon emissions is for manufacturers and suppliers to invest in improving all aspects of their supply chains, such as choosing low-carbon raw materials and reducing utility use, including investing in digital technologies [41]. Factors influencing firms to invest in carbon-reducing technologies include consumer preference for low-carbon products and government regulations [42]. However, as consumers and governments outside the supply chain cannot accurately know the carbon emissions generated by product production and supply chain members need to obtain accurate information related to carbon emissions, low-carbon supply chains need to facilitate transparency of the trading process and real-time information sharing [43,44]. In this context, carbon footprint (CF) traceability can increase the transparency of carbon emissions and build trust with external regulators [44]. In addition, to improve brand image, the CF can be reported to a third party or disclosed to the public to encourage consumers to purchase products that provide carbon footprint information [45]. Traditional approaches to carbon footprint traceability include carbon footprint inventories and automated electronic data capture, such as IOT technology. However, the above method has low efficiency of information exchange and cannot guarantee that the information is unchangeable; it requires a new traceability mechanism to achieve transparency, credibility, and fast tracking between the supply chain's upstream and downstream. As a tool for storing, monitoring, tracking, and managing the carbon emissions of participants throughout the product lifecycle, blockchain technology can be used to track the CF of different stakeholders in the complex supply chain network [46].

By integrating the CF into the blockchain, the carbon emission information of related products can be searched, and the sharing of carbon emission information across the supply chain can be realized, thus forming a consensus on carbon emission reduction. The integration layer is used in the blockchain to retrieve the carbon footprint information of each enterprise and, when the enterprise needs to calculate the carbon footprint, it can be searched when a component's carbon footprint is recorded on the blockchain or reported in a database. A consensus on reducing carbon emissions can be formed by incorporating the carbon footprint into the blockchain, which allows for the sharing of carbon emission data along the supply chain and the search of carbon emission information for related products [47]. Each enterprise's carbon footprint is retrieved from the blockchain via the integration layer, which can also be searched when a component's carbon footprint is reported in a database or is recorded on the blockchain; the enterprise needs to compute its carbon footprint [48]. Some scholars have proposed a supply chain environmental analysis tool to evaluate the carbon emissions of each entity involved in the product lifecycle in the supply chain. It combines blockchain with IOT, AI, and machine learning technologies to reduce carbon emissions and meet the needs of buyers [49]. As a result, researchers have studied blockchain-based carbon emission tracing in great detail: (1) Developed the carbon footprint chain, a cluster-based blockchain implementation technique that provides

a low-cost, distributed record-keeping system to track the CF of food transportation while maintaining privacy [50]. (2) Blockchain technology adoption in the construction industry can help the sector participate in the carbon credit market and create a precise and safe measurement, reporting, and verification (MRV) system for energy consumption and carbon emissions required by climate change projects [51].

The blockchain technology application in product carbon emission traceability can avoid carbon emission data fraud, cultivate market trust in products and PS, allow consumers to track the production chain of products, verify their sources, and obtain quantification of the carbon footprint caused by products in the environment [13]. However, there are problems with companies in the supply chain using only the blockchain to process data: (1) Limited information stored on the chain; (2) In the scenario of a large number of accesses, timely traceability queries are more complex; (3) The information processing capacity is poor. Therefore, based on social technology theory (STT), blockchain technology is considered to be an exogenous structure that needs to integrate enterprises in the supply chain to achieve the purpose of using blockchain as a strategic tool for carbon reduction [35]. Among solutions for the use of blockchain technology in the supply chain of raw materials, regulatory and market requirements for carbon traceability are being monitored and audited by government agencies, monitoring and evaluating blockchain implementations in the low-carbon commodity sector and adjusting carbon allowances and green technology adoption subsidies accordingly. Therefore, in the establishment of a blockchain traceability system for the low-carbon commodity supply chain, the government has a leadership, oversight, and decision-making role to play in facilitating blockchain technology adoption by businesses, including PMs and PSs [8].

*2.4. Evolutionary Game Theory*

Evolutionary game theory is a mathematical method for the study and prediction of social interactions and a theory that combines game-theoretic analysis with the analysis of dynamic evolutionary processes. It assumes that participants are boundedly rational, that the equilibrium is the result of continuous adaptation and improvement rather than one-time selection, and that it shifts even when stability is reached [52]. Similar to the classical Nash equilibrium, there is an ESS. When a state can be maintained despite small perturbations caused by a dynamic system, it is said to be a steady state. In addition to the concept of evolutionarily stable strategies, evolutionary game theory also considers replicator dynamics. The replicator dynamics model can better predict the trend of individual strategy selection in populations [53].

The supply chain comprises a complex network of multiple member companies. Upstream and downstream companies can benefit from the traceability of the carbon footprint of the production process of goods. When making strategic choices about adopting blockchain technology, member firms often lack sufficient information. EGT is a mathematical method for analyzing the strategic choices of a large number of stakeholder actors. EGT has advantages for studying blockchain technology adoption in low-carbon supply chains. The conclusions drawn are instructive for promoting carbon emissions traceability based on blockchain technology. Several studies have employed the evolutionary game model in a two-level green supply chain composed of green suppliers and green manufacturers to investigate the internal and external factors that influence the behavior of both sides of the game but have not touched on the role of government supervision in the green supply chain.

## 3. Evolutionary Game Model

*3.1. Game Model Assumption*

3.1.1. Model Hypothesis

1.  All stakeholders aim to maximize their own interests and make strategic decisions based on finite rationality. They have the option to adopt blockchain traceability.
2.  The initial proportion of stakeholders choosing traceability strategies does not impact the final outcome.

3. Changes in various parameters will affect the decision making of corresponding stakeholders, which will ultimately be reflected in the speed of evolution and the results.
4. Rewards and penalties do not have equal effectiveness for all stakeholders.

### 3.1.2. Stakeholders

Product manufacturers. In this research, PMs are the stakeholders responsible for manufacturing and processing products. We adopt the assumption that PMs choose their traceability strategy primarily based on the overall benefits that can be obtained using various approaches and that their primary consideration when making decisions is maximizing their own profitability.

Product suppliers. The PSs refer to the manufacturing suppliers who are responsible for connecting the PMs with raw materials, establishing raw material distribution networks, and docking directly with the market. It is assumed that PSs can choose between two strategies: traceability and not-traceability. Moreover, it is assumed that after the PSs purchase products from the PMs, in order to make reasonable decisions, they strictly check whether the carbon emission traceability technology is adopted in the production of goods.

Local governments. To ensure blockchain-based traceability systems operate smoothly, the LGs should work with low-carbon supply chain participants. [44]. For instance, the LGs will have mainly policy and regulatory functions. The LGs will also provide subsidies and other penalties to change the behavior of PSs and promote low-carbon supply chains [54]. As blockchain is a new technology, the government needs to be able to build new organizational forms that correspond to it and guide relevant actors to apply blockchain to achieve more responsible forms of trust [55].

Current research indicates that blockchain technology implementation across the supply chain is contingent upon factors such as consumer awareness of traceability, production costs for suppliers and manufacturers, and the expenses associated with using blockchain technology. In addition, this paper assumes that suppliers assist manufacturers in sharing abatement costs in the form of subsidies, based on the theory of supply chain synergy [56]. On the basis of the above assumptions as well as practical considerations, we set various parameters to construct a tripartite evolutionary game model. The parameters include additional brand values to indicate consumer preference for traceability of carbon footprints [57].

### 3.1.3. Parameter Assumption

Combined with the actual situation of the blockchain implementation process, we determined the strategic parameters of the three main bodies without losing the premise of universality. The parameter settings are specifically described in Table 2.

**Table 2.** Parameters and descriptions.

| Parameters | Descriptions | Notes |
|---|---|---|
| $R_h, R_p$ | Benefits of the PMs and the PSs when choosing the traceability strategy | $R_h, R_p \geq 0$ |
| $R_l, R_n$ | Benefits of the PMs and the PSs when not choosing the traceability strategy | $R_l, R_n \geq 0$ |
| $C_h, C_p$ | Cost of choosing the traceability strategy for the PMs and the PSs | $C_h, C_p \geq 0$ |
| $Q_l, Q_p$ | Free-riding benefits of the PMs the PSs not choosing the traceability strategy while the PSs adopting | $Q_l, Q_p \geq 0$ |
| $S_h$ | Subsidies of the PMs from the PSs for choosing the traceability strategy | $S_h \geq 0$ |
| $F_n$ | Penalties of the PMs from the PSs for not choosing the traceability strategy | $F_n \geq 0$ |
| $S_o$ | Additional brand value of the PMs choosing the traceability strategy when the LGs strictly regulate | $S_o \geq 0$ |
| $S_p$ | Subsidies of the PSs from the LGs for choosing the traceability strategy | $S_p \geq 0$ |
| $F_l$ | Penalties of the PSs from the LGs for not choosing the traceability strategy | $F_l \geq 0$ |
| $G_h$ | Utilities of the LGs when the PSs adopt the strict regulation strategy | $G_h > G_l$ |
| $G_l$ | Utilities of the LGs when the PSs adopt the passive regulation strategy | |
| $C_g$ | Cost of the LGs when strictly regulating | $M > C_g$ |
| $M$ | Additional benefit of the LGs when strictly regulating | |
| $U$ | negative benefits of the LGsWhen negative regulation causes the PMs and the PSs to not choose the traceability strategy | $U \geq 0$ |

Note: Based on the factual circumstances in China, it is presumed that $G_h > G_l$. Since the government is really the main player in carbon footprint traceability, this analysis makes the assumption that $M > C_g$.

### 3.2. Replicator Dynamic of the Game Model

After the determination of the game strategies of the three subjects, the probability that the manufacturer chooses the traceability strategy is $x$, and the probability that the manufacturer does not choose the traceability strategy is $1 - x$. The probability that the supplier chooses the traceability strategy is $y$, and the probability that the supplier chooses "not traceability" is $1 - y$. The probability that the government strictly regulates is $x$, and the probability that it passively regulates is $1 - z$. The tripartite game benefits of the PMs, the PSs, and the LGs under various behavioral strategies are shown in Table 3.

**Table 3.** The payoffs of the LGs, the PMs, and the PSs.

| PMs \ PSs | **The LGs Choose the Strict Regulation Strategy** | |
|---|---|---|
| | traceability | not traceability |
| traceability | $(R_h + S_h + S_o - C_h, R_p + S_p - S_h - C_p, G_h + M - C_g - S_p)$ | $(R_h + S_h + S_o - C_h, R_n + Q_p - F_l - S_h, G_l + F_l + M - C_g)$ |
| not traceability | $(R_l + Q_l - F_n, R_p + F_n - C_p, G_h + M - C_g)$ | $(R_l - F_n, R_n + F_n - F_l, G_l + M + F_l - C_g)$ |
| PMs \ PSs | The LGs choose the passive regulation strategy | |
| | traceability | not traceability |
| traceability | $(R_h + S_h - C_h, R_p + S_p - C_p - S_h, G_h - S_p)$ | $(R_h + S_h - C_h, R_p + Q_p - S_h, G_l)$ |
| not traceability | $(R_l + Q_l - F_n, R_p + F_n - S_h, G_h)$ | $(R_l - F_n, R_p + F_n, G_l - U)$ |

#### 3.2.1. The PMs' Anticipated Rewards and Strategy Analysis

According to Table 3, the expected payoffs can be calculated if the PM chooses traceability $E_{1h}$ or "not-traceability" $E_{1l}$. Then, the average expected payoff of the PMs is recorded as $E_1$.

$$E_{1h} = yz(R_h + S_h + S_o - C_h) + y(1 - z)(R_h + S_h - C_h) + (1 - y)z(R_h + S_h + S_o - C_h) \\ + (1 - y)(1 - z)(R_h + S_h - C_h) \tag{1}$$

$$E_{1l} = yz(R_l + Q_l - F_n) + y(1 - z)(R_l + Q_l - F_n) + (1 - y)z(R_l - F_n) + (1 - y)(1 - z)(R_l - F_n) \tag{2}$$

$$E_1 = xE_{1h} + (1 - x)E_{1l} \tag{3}$$

According to Equations (1)–(3), the dynamics of the replicator using the traceability of the PMs are determined as follows:

$$F_{(x)} = dx/dt = x(E_{1_h} - E_1) = x(1 - x)(R_h + S_h + F_n + zS_o - C_h - R_l - yQ_l). \tag{4}$$

Let $I_{(y)} = R_h + S_h + F_n + zS_o - C_h - R_l - yQ_l$. Then, it can be simplified as $F_1(x) = x(1 - x)I_1(y)$ and $dH_{1(x)}/dx = (1 - 2x)I_1(y)$.

When $y = (R_h + S_h + F_n + zS_o - C_h - R_l)/Q_l = y_*$, $I_{(y)} = 0$; at this point, $F_1(x) = 0$. Thus, whatever the initial ratio of "traceability" x to "not traceability" $1 - x$ is, this ratio does not change over time.

According to the stability theorem for differential equations, the evolutionary stabilization strategy satisfies the following: $F_1(x) = 0$ and $\partial F_1(x)/\partial x < 0$. As $I_1(y)/\partial y = -Q_l < 0$, when $y < y_*$, $I_1(y) > 0$, $\partial F(x)/\partial x|_{x=1} < 0$, and $\partial F_1(x)/\partial x|_{x=0} > 0$; consequently, $x = 1$ is the evolutionary stability strategy (ESS). When $y > y_*$, $I_1(y) < 0$, $\partial F(x)/\partial x|_{x=0} < 0$, and $\partial F_1(x)/\partial x|_{x=1} > 0$, meaning $x = 0$ is ESS.

#### 3.2.2. The PSs' Anticipated Rewards and Strategy Analysis

Then, the expected payoffs can be calculated if the PSs chooses "traceability" $E_{2p}$ or "not traceability" $E_{2n}$. Then, the PSs' average projected payment is noted as $E_2$.

$$E_{2p} = xz(R_p + S_p - C_p - S_h) + x(1 - z)(R_p + S_p - C_p - S_h) + z(1 - x)(R_p + F_n - C_p) \\ + (1 - x)(1 - z)(R_p + F_n - C_p) \tag{5}$$

$$E_{2n} = xz(R_n + Q_p - F_l - S_h) + x(1 - z)(R_n + Q_p - S_h) + z(1 - x)(R_n + F_n - F_l) \\ + (1 - x)(1 - z)(R_n + F_n) \tag{6}$$

$$E_2 = yE_{2p} + (1-y)E_{2n} \tag{7}$$

According to Equations (5)–(7), the dynamics of the replicator using the traceability of the PSs are determined as follows:

$$F_2(y) = \frac{dy}{dt} = y(E_{2p} - E_2) = y(1-y)\left(R_p - C_p - R_n + x\left(S_p - Q_p\right) + zF_l\right). \tag{8}$$

Let $I_2(z) = R_p - C_p - R_n + x\left(S_p - Q_p\right) + zF_l$. Then, $F_{2(y)} = y(1-y)$ and $\partial F_2(y)/\partial y = (1-2y)I_{2(z)}$. When $z = \left(R_n + C_p - R_p - x\left(S_p - Q_p\right)\right)/F_l = z_*$, $I_2(z) = 0$; at this point, $H_2(y) = 0$.

As $\partial I_2(z)/\partial z = F_l > 0, z < z*, I_2(z) < 0, \partial F_2(y)/\partial y \mid_{y=0} < 0$, and $\partial F_2(y)/\partial y \mid_{y=1} > 0$, meaning $y = 0$ is ESS; when $z > z_*$, $I_2(z) > 0, \partial F_2(y)/\partial y \mid_{y=0} > 0, \partial F_2(y)/\partial y \mid_{y=1} < 0$. So, $y = 1$ is ESS.

### 3.2.3. The LGs' Anticipated Rewards and Strategy Analysis

Finally, the expected payoffs can be calculated if the LGs chooses "strict regulation" $E_{3r}$ or "negative regulation" $E_{3o}$. Then, the average expected payoff of the PSs is recorded as $E_2$.

$$E_{3r} = xy\left(G_h + M - C_g - S_p\right) + x(1-y)\left(G_l + F_l + M - C_g\right) + y(1-x)\left(G_h + M - C_g\right) \\ + (1-x)(1-y)\left(G_l + F_l + M - C_g\right) \tag{9}$$

$$E_{3o} = xy\left(G_h - S_p\right) + x(1-y)(G_l) + y(1-x)(G_h) + (1-x)(1-y)(G_l - U) \tag{10}$$

$$E_3 = zE_{3r} + (1-z)E_{3p} \tag{11}$$

The replicator dynamics for the strict regulation of the local LGs are given by Equations (9)–(11), as follows:

$$F_3(z) = \frac{dz}{dt} = z(E_{3r} - E_3) = z(1-z)(M + F_l + U - Cg - yF_l - xU - yU + xyU). \tag{12}$$

Let $I_3(x) = M + F_l + U - Cg - yF_l - Xu - Yu + xy$. Then, it can be rewritten as $F_3(z) = z(1-z)I_3(x)$ and $dF_3(z)/dz = (1-2z)I_3(x)$.

When $x = (M + F_l + U - Cg - yF_l + yu)/-(1-y)U = x_*$, $I_3(x) = 0$ and $F_3(z) == 0$.

Thus, whatever the starting ratio of "strict regulation" $z$ to "passive regulation "$1-z$", this ratio does not change over time. As $\partial I_3(x)/\partial x = -(1-y)U < 0$, when $x < x*$ $I_3(x) > 0, \partial F(z)/\partial z \mid_{z=0} > 0$ and $F(z)/\partial z \mid_{z=1} < 0$; it means $z = 0$ is ESS. When $x > x_*$, $I_3(x) > 0, \partial F_3(z)/\partial z \mid_{z=0} < 0$, and $\partial F_3(z)/\backslash\partial z \mid_{z=1} > 0$; so, $z = 1$ is ESS.

## 4. Equilibrium Points and Stability Analysis

The tripartite system's overall analysis is carried out based on the stability analysis of the PMs', the PSs' and the LGs' strategies.

$$\begin{cases} F_1(x) = dx/dt = x(E_{1h} - E_1) = x(1-x)(R_h + S_h + F_n + zS_o - C_h - R_l - yQ_l) \\ F_2(y) = dy/dt = y(E_{2p} - E_2) = y(1-y)\left(R_p - C_p - R_n + x\left(S_p - Q_p\right) + zF_l\right) \\ F_3(z) = dz/dt = z(E_{3r} - E_3) = z(1-z)(M + F_l + U - Cg - yF_l - xU - yU + xyU) \end{cases} \tag{13}$$

When $\frac{dx}{dt} = 0$, $\frac{dy}{dt} = 0$ and $\frac{dz}{dt} = 0$. From Equation (13), the equilibrium points of the system can be given: $E_1(0,0,0), E_2(1,0,0), E_3(0,1,0), E_4(0,0,1), E_5(1,1,0), E_6(1,0,1), E_7(0,1,1)$, and $E_8(1,1,1)$. Morever, mixed-strategy equilibrium points $E*_{9-12}$ can be given as follows:

$$E_9^* = \left(\left(R_n + C_p - R_p\right)/\left(S_p - Q_p\right), (R_h + F_n + S_h - R_l - C_h)/Q_l, 0\right),$$
$$E_{10}^* = \left(0, \left(M + F_l + U - C_g\right)/(F_l + U), \left(R_n + C_p - R_p\right)/F_l\right),$$
$$E_{11}^* = \left(1, \left(M + F_l - C_g\right)/F_l, \left(R_n + C_p + Q_p - R_p - S_p\right)/F_l\right),$$
$$E_{12}^* = \left(\left(R_n + C_p - R_p - F_l\right)/\left(S_p - Q_p\right), (R_n + S_h + F_n + S_o - C_h - R_l)/Q_l, 1\right).$$

However, if the equilibrium point in an asymmetric game is asymptotically stable, it must be consistent with a rigorous Nash equilibrium and a pure strategy equilibrium [48]. Therefore, it is sufficient to analyze the equilibrium point of a pure strategy replicated dynamic equation in order to obtain the asymptotic stability of the equilibrium point of a replicated dynamic equation. The requirements are satisfied at the following equilibrium locations: the Liapunov system stability criterion states that an equilibrium point is unstable if one or more $\lambda > 0$ and asymptotically stable if all eigenvalues of the Jacobi matrix are $\lambda < 0$ [50]. Equation (14) provides the Jacobi matrix J.

$$J = \begin{bmatrix} \frac{\partial F_1(x)}{\partial x} & \frac{\partial F_1(x)}{\partial y} & \frac{\partial F_1(x)}{\partial z} \\ \frac{\partial F_2(y)}{\partial x} & \frac{\partial F_2(y)}{\partial y} & \frac{\partial F(y)}{\partial z} \\ \frac{\partial F_3(z)}{\partial x} & \frac{\partial F_3(z)}{\partial y} & \frac{\partial F_3(z)}{\partial z} \end{bmatrix} \tag{14}$$

$$= \begin{bmatrix} (1-2x)(R_h + S_h + F_n + zs_o - R_l - C_h - yQ_l) & -x(1-x)Q_l & x(1-x)S_o \\ y(1-y)\left(S_p - Q_p\right) & (1-2y)\left(R_p + xSp + zFl - xQ_p - Rn - Cp\right) & y(1-y)F_l \\ -z(1-z)(1-y)U & -z(1-z)(F_l + U - xU) & (1-2z)(M + Fl + U - Cg - yFl - xU - yU + xyU) \end{bmatrix}$$

Table 4 shows the eigenvalues of points $E_1 - E_8$.

**Table 4.** Eigenvalue of Jacobian matrix.

| Eigenvalues | | Eigenvalues | |
| --- | --- | --- | --- |
| | $\lambda_1$ | $\lambda_2$ | $\lambda_3$ |
| $E_1$ | $R_h + S_h + F_n - C_h - R_l$ | $R_p - R_n - C_p$ | $M + F_l + U - C_g$ |
| $E_2$ | $-(R_h + S_h + F_n - C_h - R_l)$ | $R_p + S_p - R_n - C_p - Q_p$ | $M + F_l - C_g$ |
| $E_3$ | $R_h + S_h + F_n - C_h - R_l - Q_l$ | $-\left(R_p - R_n - C_p\right)$ | $M - C_g$ |
| $E_4$ | $R_h + S_h + F_n + S_o - C_h - R_l$ | $R_p + F_l - R_n - C_p$ | $-(M + F_l + U - C_g)$ |
| $E_5$ | $-(R_h + S_h + F_n - C_h - R_l - Q_l)$ | $-\left(R_p + S_p - R_n - C_p - Q_p\right)$ | $M - C_g$ |
| $E_6$ | $-(R_h + S_h + F_n + S_o - C_h - R_l)$ | $R_p + S_p + F_l - R_n - C_p - Q_p$ | $-(M + F_l - C_g)$ |
| $E_7$ | $R_h + S_h + F_n + S_o - C_h - R_l - Q_l$ | $-\left(R_p + F_l - R_n - C_p\right)$ | $-\left(M - C_g\right)$ |
| $E_8$ | $-\left(R_h + S_h + F_n + S_o - C_h - R_l - Q_l\right)$ | $-\left(R_p + S_p + F_l - R_n - C_p - Q_p\right)$ | $-\left(M - C_g\right)$ |

Through observation, we can observe that the eigenvalues $\lambda_3$ of the $E_1(0, 0, 0)$, $E_2(1, 0, 0)$, $E_3(0, 1, 0)$, and $E_5(1, 1, 0)$ are positive under the premise that $M > C_g$. Therefore, these equilibria are not asymptotically stable points, so only $E_4(0, 0, 1)$, $E_6(1, 0, 1)$, $E_7(0, 1, 1)$, and $E_8(1, 1, 1)$ need to be examined; Table 5 shows the stability conditions.

**Table 5.** System equilibrium stability conditions.

| Equilibrium Points | Stability Conditions | Scenario |
| --- | --- | --- |
| $E_4$ | $R_h + S_h + F_n + S_o - C_h - R_l < 0;$<br>$R_p + F_l - R_n - C_p < 0;$<br>$-\left(M + F_l + U - C_g\right) < 0$ | 1 |
| $E_6$ | $-(R_h + S_h + F_n + S_o - C_h - R_l) < 0;$<br>$R_p + S_p + F_l - R_n - C_p - Q_p < 0;$<br>$-\left(M + F_l - C_g\right) < 0$ | 2 |
| $E_7$ | $R_h + S_h + F_n + S_o - C_h - R_l - Q_l < 0;$<br>$-\left(R_p + F_l - R_n - C_p\right) < 0;$<br>$-\left(M - C_g\right) < 0$ | 3 |
| $E_8$ | $-(R_h + S_h + F_n + S_o - C_h - R_l - Q_l) < 0;$<br>$-\left(R_p + S_p + F_l - R_n - C_p - Q_p\right) < 0;$<br>$-\left(M - C_g\right) < 0$ | 4 |

## 5. Numerical Simulation

To illustrate the process of the evolution of the four ESSs under different parameter scenarios, this paper uses MATLAB 2016 software to numerically simulate the evolutionary game process of the PMs, the PSs, and the LGs. And, by quantitatively changing the value, the influence of key parameters in the replication dynamics on the process and results of the tripartite evolutionary game is more intuitive.

### 5.1. ESSs in Different Scenarios

Since the system has multiple evolutionary paths, we selected three groups of four equilibrium points adopting pure strategies, including $E_4$, $E_6$, $E_7$, and $E_8$, to perform numerical simulations to investigate each stakeholder's evolutionary process and confirm the accuracy and applicability of the model using the evolutionary stability test results. And, given the initial value (randomly taken between 0 and 1), Equation (13) is simulated using MATLAB to obtain the evolutionary paths of the PMs, the PSs, and the LGs. In order to ensure the rationality of the original parameter setting, the model parameters must satisfy the economic assumptions and empirical determination. According to the practical significance of the model parameters and the previous research experience, the parameter settings are shown in Table 6.

**Table 6.** Simulation data in four scenarios.

|  | $R_h$ | $R_l$ | $C_h$ | $S_h$ | $Q_l$ | $F_l$ | $R_p$ | $R_n$ | $C_p$ | $S_p$ | $Q_P$ | $F_n$ | M | $C_g$ | U | $S_o$ |
|---|---|---|---|---|---|---|---|---|---|---|---|---|---|---|---|---|
| Scenario 1 | 10 | 25 | 15 | 10 | 10 | 8 | 32 | 28 | 20 | 15 | 5 | 12 | 20 | 10 | 10 | 5 |
| Scenario 2 | 15 | 20 | 12 | 20 | 16 | 4 | 32 | 24 | 25 | 10 | 10 | 5 | 20 | 10 | 10 | 5 |
| Scenario 3 | 10 | 15 | 15 | 8 | 20 | 18 | 30 | 28 | 20 | 15 | 10 | 8 | 20 | 10 | 10 | 5 |
| Scenario 4 | 20 | 15 | 15 | 10 | 10 | 10 | 28 | 20 | 18 | 18 | 5 | 10 | 20 | 10 | 10 | 5 |

### 5.1.1. Scenario 1

According to the parameters in Table 6, the asymptotically stable condition is satisfied at $E_4(0, 0, 1)$, that is $R_h + S_h + S_o - C_h < R_l - F_n$ and $R_p - C_p < R_n - F_l$. Assuming $x_o = 0.2$, $y_o = 0.4$ and $z_0 = 0.5$; since the PMs are under pressure to invest in hardware equipment upfront and the PSs need to migrate existing product information, the LGs will actively perform their duties so long as there is a greater net benefit from stringent government regulation than there is from negative regulation. The PMs and the PSs will not choose the traceability strategy, by taking into account that the advantages of the PSs and PMs selecting the "not traceability" strategy outweigh the advantages of traceability. The simulation results are shown in Figure 2. The eigenvalues of the equilibrium points $E_4(0, 0, 1)$ are all subzero, so $E_4(0, 0, 1)$ is the evolutionary stability point. The simulation findings show that while the probability/proportion of the LGs carrying out their responsibilities increases the probability/proportion of the PMs and PSs choosing a traceability approach drops as the iteration process goes on. At this point, the evolutionary game strategy combination of the PMs, the PSs, and the LGs is {no traceability, no traceability, and strict regulation}. When the above conditions are met, for the LGs, the regulation to promote the implementation of traceability technology is ineffective, as demonstrated by the tripartite evolutionary game's outcomes.

### 5.1.2. Scenario 2

The experiment displays the findings for each of the 125 initial value groups. In Figure 3a, the final evolutionary results of the random initial proportion combination are uniform, and the evolutionary stability verifies the correctness of the model. The asymptotically stable condition is satisfied at $E_6(1, 0, 1)$, that is $R_h + S_h + S_o - C_h > R_l - F_n$ and $R_p + S_p - C_p < R_n + Q_p - F_l$. The eigenvalues of $\lambda_1$, $\lambda_2$, and $\lambda_3$ to the equilibrium point $E_6 (1, 0, 1)$ are all less than zero. The simulation results (Figure 3) show that, as the iteration process moves forward, the probability/proportion of the PSs choosing a traceability

strategy decreases, while the probability/proportion of the PMs choosing a traceability strategy and the probability/proportion of the LGs strictly regulating increase. At this point, the evolutionary game strategy combination of the PMs, the PSs, and the LGs is {traceability, not traceability, and strict regulation}. From the first inequality $R_h + S_h + S_o - C_h > R_l + F_l$, it is evident that the advantages of implementing the "traceability" strategy outweigh the advantages of implementing the "no traceability" strategy and the manufacturer will adopt the blockchain traceability. From the second inequality $R_p + S_p - C_p < R_n + Q_p - F_l$, it can be seen that when the manufacturer adopts the "traceability" strategy, as long as the benefit of the supplier's "not traceability" strategy exceeds the return of the "traceability" strategy, it will not take the "traceability" strategy. However, the total benefit of the LGs' strict regulatory strategy is still greater than that of negative regulation, and the LGs will still strictly regulate the traceability of carbon emissions from low-carbon commodities. In this scenario, the parameter's values are shown in Scenario 2 in Table 6, which meet the local equilibrium point $E_6$, and in Figure 3 of the evolutionary path.

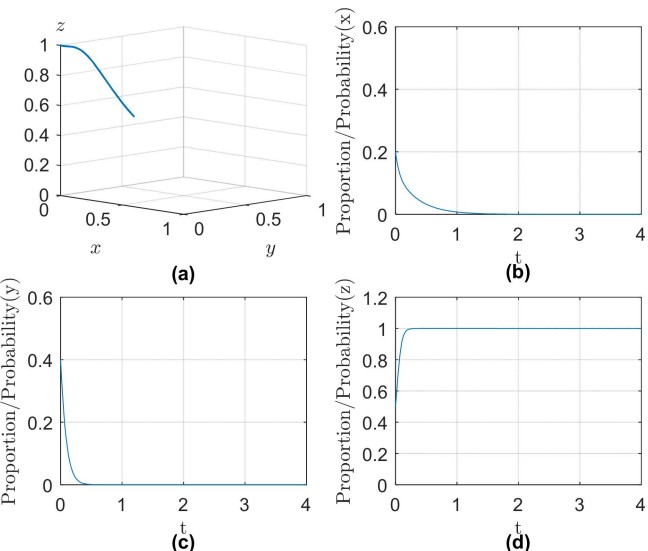

**Figure 2.** Paths of evolution for all players (**a**), the PMs (**b**), the PSs (**c**), and the LGs (**d**) when they are evolving toward the stable equilibrium point $E_4$ (0, 0, 1).

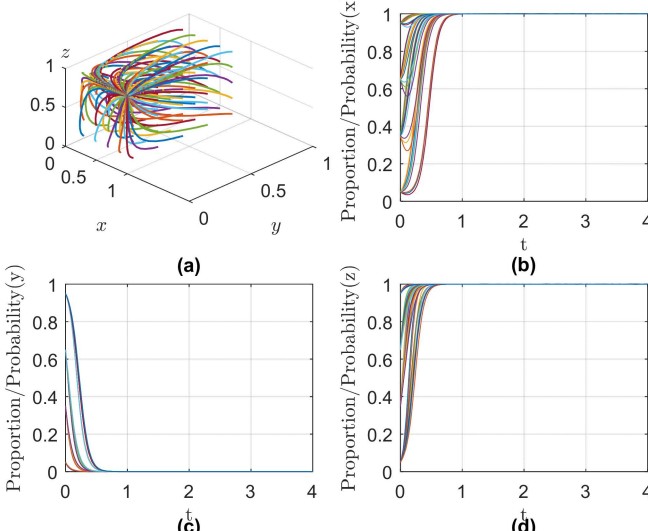

**Figure 3.** Paths of evolution for all players (**a**), the PMs (**b**), the PSs (**c**), and the LGs (**d**) when they are evolving toward the stable equilibrium point $E_6$ (1, 0, 1). The curves of different colors represent evolutionary paths with different initial scales.

### 5.1.3. Scenario 3

The asymptotically stable condition is satisfied at $E_7(0, 1, 1)$, that is $R_h + S_h + S_o - C_h < R_l + Q_l - F_n$ and $R_p - C_p > R_n - F_l$. All of the eigenvalues ($\lambda_1$, $\lambda_2$, and $\lambda_3$) are less than zero to the equilibrium points $E_7(0, 1, 1)$. As the iteration process proceeds, the simulation results (Figure 4) indicate that the probability/proportion of the PSs choosing the traceability strategy and the probability/proportion of the LGs strictly regulating increases, while the probability/proportion of the PMs choosing traceability strategy decreases. After the introduction of the blockchain technology traceability system into the supply chain, the stable evolution strategy of the PMs, the PSs, and the LGs is {not traceability, traceability, and strict regulation}. For the first inequality $R_h + S_h + S_o - C_h < R_l + Q_l - F_n$, it can be found that, under the premise of strict the LGs regulation and supplier "traceability" strategy, the total return of the manufacturer's hitch-hiking strategy is higher than the total return of the "traceability" strategy, so the manufacturer chooses the "no-traceability" strategy. The second inequality $R_p - C_p > R_n - F_n$ also shows that the total yield of the PSs after adopting "traceability" exceeds the benefits after no-traceability minus punishment, so the supplier chooses the "traceability" strategy. Figure 4 shows the evolution trajectory.

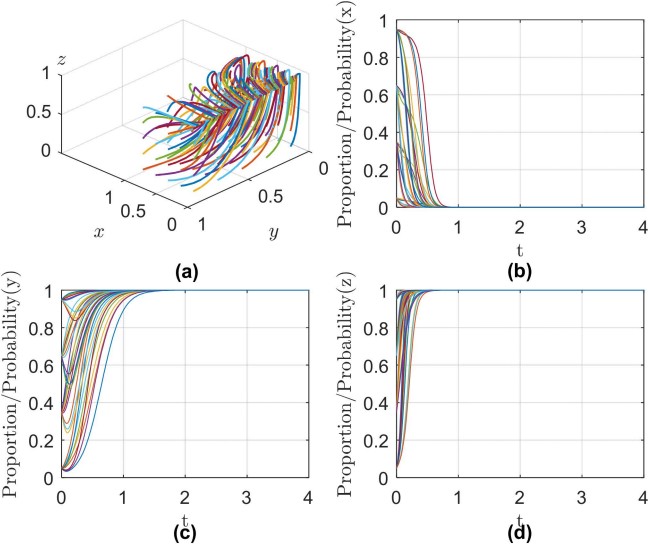

(a) (b) (c) (d)

**Figure 4.** Paths of evolution for all players (**a**), the PMs (**b**), the PSs (**c**), and the LGs (**d**) when they are evolving toward the stable equilibrium point $E_7$ (0, 1, 1). The curves of different colors represent evolutionary paths with different initial scales.

### 5.1.4. Scenario 4

The asymptotically stable condition is satisfied at $E_8(1, 1, 1)$, that is $R_h + S_h + S_o - C_h > R_l + Q_l - F_n$ and $R_p + S_p - C_p > R_n + Q_p - F_l$. All of the eigenvalues ($\lambda_1$, $\lambda_2$, and $\lambda_3$) are less than zero to the equilibrium points $E_8(1, 1, 1)$. According to the simulation results (Figure 5), as the iteration process continues, the probability/proportion of the PMs and the PSs that choose the traceability strategy and the probability/proportion of the LGs strictly regulating increase. At this moment, the stable evolutionary strategy of the PMs, the PSs, and the LGs is {traceability, traceability, and strict regulation}. As can be seen from the first inequality, when the total return of selecting the "traceability" behavior exceeds the total return of choosing the "not traceability" behavior, the supplier chooses the "traceability" strategy. Similarly, when the total return of traceability exceeds the return of non-traceability the supplier chooses the "traceability" strategy. The value in scenario 4 (Table 6) meets the local stable equilibrium point $E_7$, and Figure 5 shows the evolution trajectory. In accordance with the evolutionary trajectory, it is obvious that this is the expected effect of the LGs' supervision, and it is also in line with the economic benefits of enterprises among supply chain members. Therefore, Scenario 4 is also an ideal state of stability.

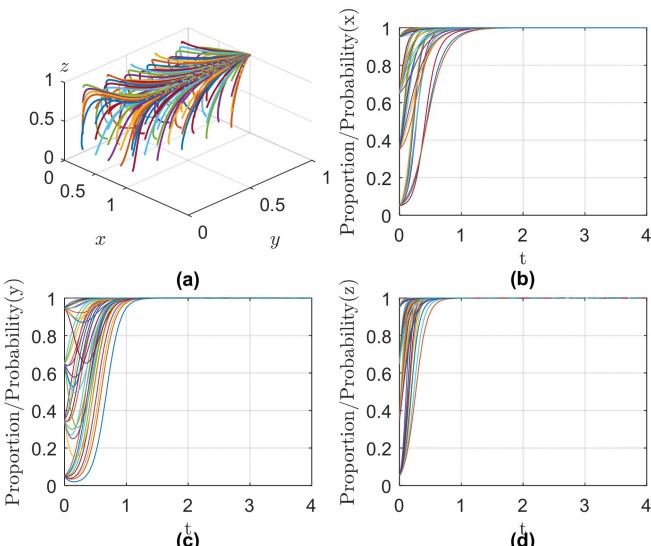

**Figure 5.** Paths of evolution for all players (**a**), the PMs (**b**), the PSs (**c**), and the LGs (**d**) when they are evolving toward the stable equilibrium point $E_8$ (1, 1, 1). The curves of different colors represent evolutionary paths with different initial scales.

### 5.2. Impacts of Parameter Variations on the Evolutionary Results

Four local equilibrium points exist in the system when three stakeholders employ pure strategies. Four of these equilibrium points have the potential to become ESSs under specific circumstances. For example, $E_8$ (1, 1, 1) is the perfect ESS. The outcomes of the four previous cases' evolutionary simulations demonstrate that only the convergence speed is impacted by the starting parameters, not the evolutionary outcomes. We do not address here how the starting state affects the system's evolution. Our research focuses on how the simulation parameters affect the results, including $R_h, R_p, C_h, C_p, Q_l, Q_p$, and $S_o$.

Now, initialize the parameter value as follows:

$$R_h = 15, R_l = 15, C_h = 15, S_h = 8, Q_l = 10, F_l = 10, R_p = 32, R_n = 18,$$
$$C_p = 18, S_p = 18, Q_p = 5, F_n = 10, M = 20, C_g = 10, U = 10, S_o = 10.$$

#### 5.2.1. The Impact of Blockchain Traceability Benefits

Firstly, we investigated sensitivity to the benefits of blockchain traceability, by varying $R_h$ and $R_p$ to simulate the evolutionary paths of both the PMs and the PSs. The evolutionary paths of the PMs, when $R_h$ is taken as 10, 15, and 20, respectively, are shown in Figure 6, and the evolutionary paths of the PSs, when $R_P$ is taken as 24, 32, and 40, respectively, are shown in Figure 7. The manufacturer's evolutionary trajectory is highly sensitive to traceable gain, as shown in Figure 6. In the early phases, when the product manufacturer chooses the traceability strategy with a profit of 10, the manufacturer will not choose the traceability strategy. When the traceability gain increases to 15 the manufacturer will turn to the traceability strategy. The higher the traceability gain of the PMs have, the faster the convergence will be. Relatively speaking, a certain range of traceability gain changes will not affect the evolution of the PSs, which is shown in Figure 7. When the supplier chooses a traceability strategy with a low gain, the probability that suppliers choose the traceability strategy is lower but then grows at a faster rate. As the traceability gain increase to 20 the PSs quickly reach equilibrium.

#### 5.2.2. The Impact of Blockchain Traceability Costs

Secondly, as shown in Figures 8 and 9, the traceability cost is a critical factor that prevents both the PMs and the PSs from choosing the "traceability" strategy. The evolutionary paths of the PMs, when $C_h$ is taken as 10, 15, and 20, respectively, are shown in Figure 8,

and the evolutionary paths of the PSs, when $C_P$ is taken as 18, 24, and 30, respectively, are shown in Figure 9.

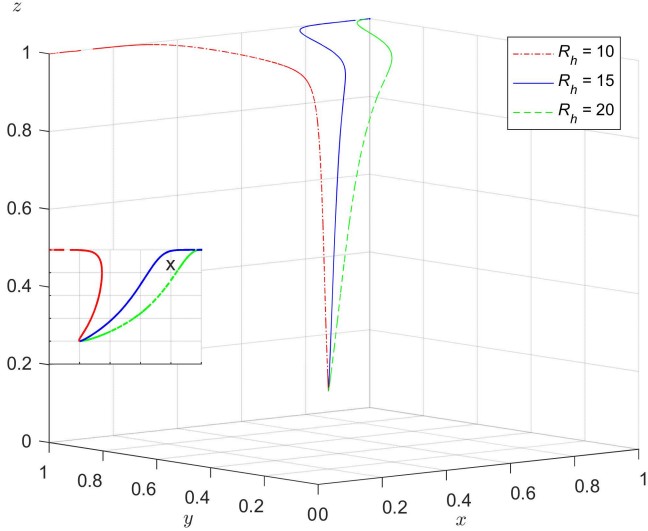

**Figure 6.** Simulation of the PMs' evolution strategies under various traceability benefits represented by different color curves.

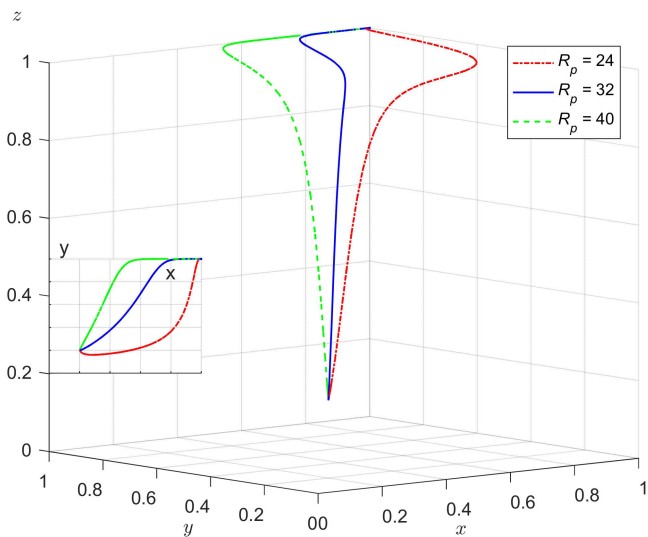

**Figure 7.** Simulation of the PSs' evolution strategies under various traceability benefits represented by different color curves.

Traceability costs include data migration costs, infrastructure investment, and resources consumed by distributed computing. All other conditions being equal, when the traceability costs to the PMs are 20 the PMs finally choose the "not traceability" strategy. However, if the cost of traceability is reduced to a certain threshold then the PMs will turn to the traceability strategy. However, due to the strict regulation of the LGs to invest in blockchain technology, the impact of significant traceability cost changes and the PS only exists when it evolves to a stable state without changing the evolutionary result. The probability of the supplier choosing a traceability strategy with a cost of 30 is low when the supplier chooses a traceability strategy, but then the probability tends to increase rapidly. In addition, the lower the cost of traceability, the less time it takes for PMs to evolve to a stable state.

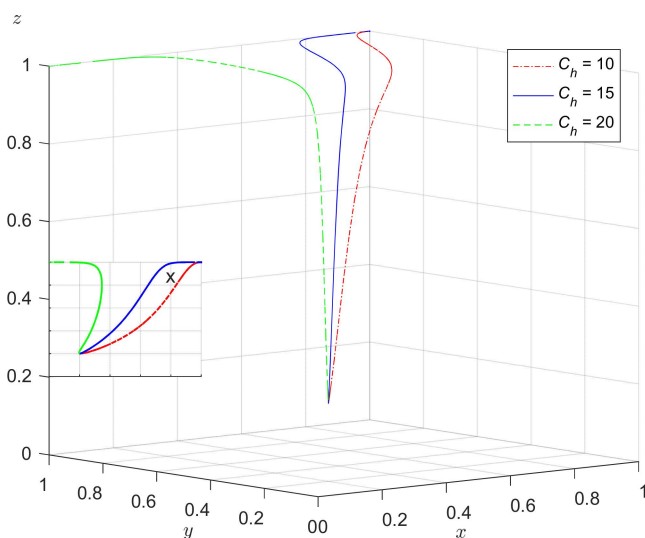

**Figure 8.** Simulation of the PMs' evolution strategies under various traceability costs represented by different color curves.

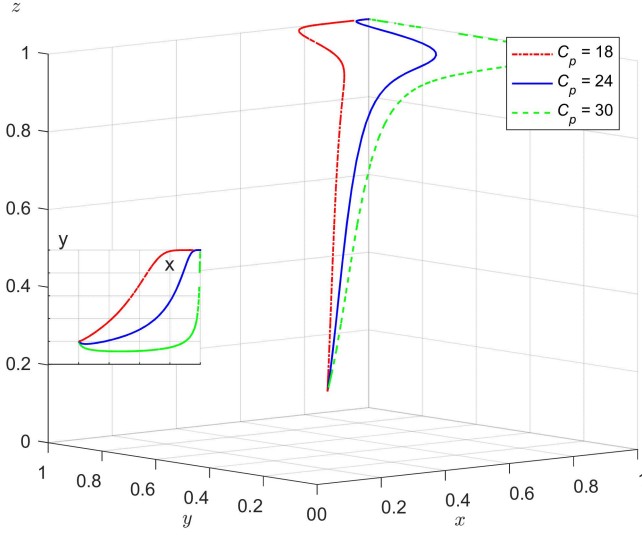

**Figure 9.** Simulation of the PSs' evolution strategies under various traceability costs represented by different color curves.

Therefore, in the initial stage, the PMs and the PSs often choose the "no traceability" strategy. However, considering the LGs' strict regulatory measures and the maturity of blockchain technology, the cost has decreased to some extent. As the cost of traceability decreases, they will gradually change their strategy and opt for traceability.

### 5.2.3. The Impact of Free-Riding Benefits

Then, we perform evolutionary path simulations for both the PMs and the PSs to investigate the influence of traceability benefit by changing $Q_l$ and $Q_p$. The evolutionary paths of the PMs, when $Q_L$ is taken as 8, 12, and 16, respectively, are shown in Figure 10, and the evolutionary paths of the PSs, when $Q_p$ is taken as 18, 24, and 30, respectively, are shown in Figure 11.

Similar to traceability costs, for the PMs and the PSs, because the PMs and the PSs are at different stages of the supply chain, the "free-riding" benefits have different effects on the PMs' and the PSs' selection strategies. The PMs and the PSs may choose different strategies, especially when the value changes significantly. In a low-carbon supply chain that contains both the PMs and the PSs, if only one party chooses the "traceability" strategy

it will positively affect the overall interests of the supply chain, resulting in a corresponding "free-riding" benefit for the other party. On the other hand, if one of the parties receives a significant free-riding benefit because the other party chooses the traceability strategy the manufacturer or supplier will tend to not choose the traceability strategy. Only if the free-riding benefit is less than some threshold value, i.e., the subject will choose the traceability strategy.

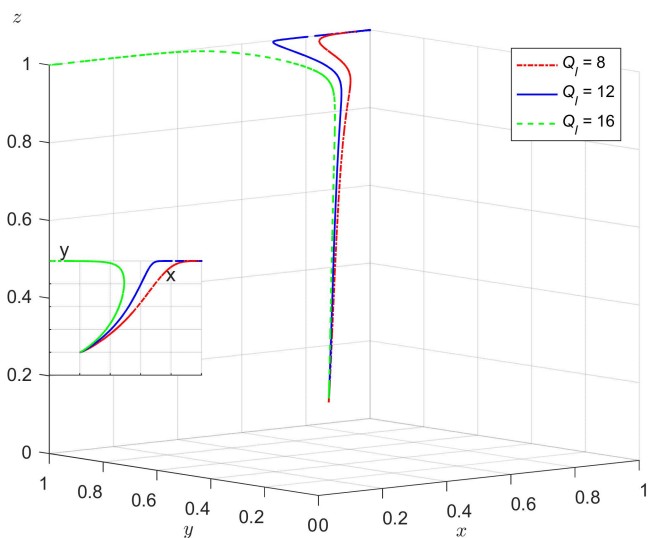

**Figure 10.** Simulation of the PMs' evolution strategies under various free-riding benefits represented by different color curves.

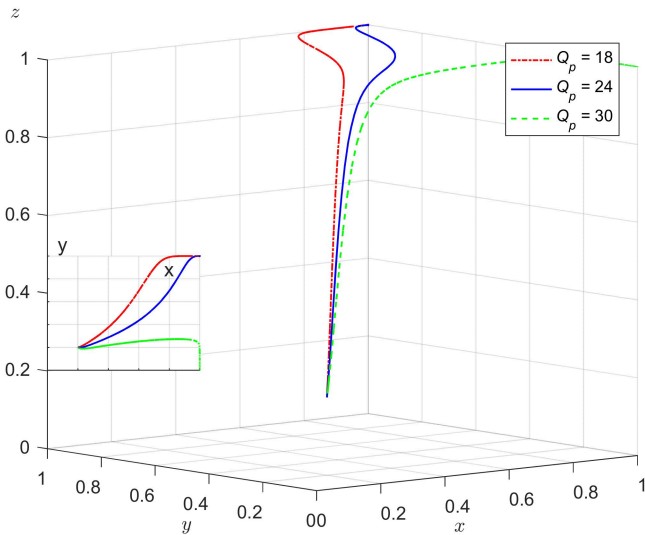

**Figure 11.** Simulation of the PSs' evolution strategies under various free-riding benefits represented by different color curves.

It also needs the intervention and regulation of the LGs to avoid hitch-hiking.

### 5.2.4. The Impact of Brand Benefits

Finally, we perform evolutionary path simulations for both the PMs and the PSs to investigate the influence of traceability benefit by changing $S_o$. The evolutionary paths of the PMs, when $S_o$ is taken to be 10, 20, and 30, respectively, are shown in Figure 12.

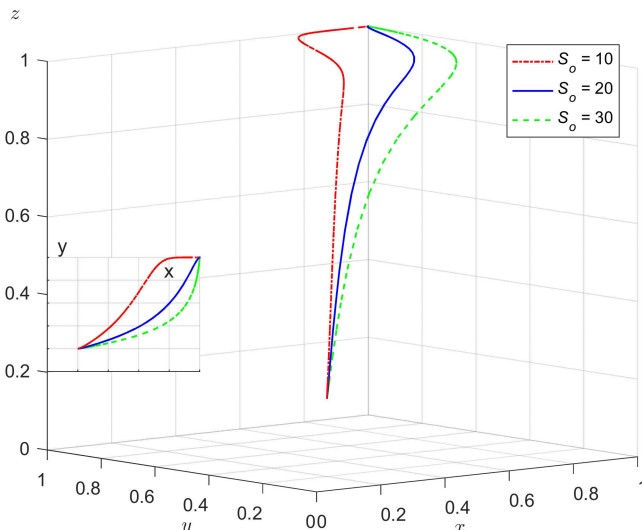

**Figure 12.** Simulation of the PMs' evolution strategies under various brand benefits represented by different color curves.

In the initial state, when the brand benefit obtained by the manufacturer choosing traceability is 10 the probability of the manufacturer choosing traceability is low, although it increases later but the speed is slow. When the traceable brand benefit increases to 20 the manufacturer quickly reaches equilibrium. Unlike the initial brand improvement, the speed of reaching equilibrium does not increase much as the traceability benefit increases from 20 to 30, as shown in Figure 12. Therefore, the PMs should increase marketing investment to build low-carbon brand attractiveness. However, when the deployment of blockchain technology enters a mature stage, marketing investment should be controlled and the appeal of flat auction should be gradually improved by relying on product word-of-mouth.

### 5.3. The Analysis of the Effectiveness of Subsidies and Penalties

Subsidies for low-carbon technologies can promote low-carbon manufacturing by member firms of the supply chain, including manufacturers [58]. To explore the extent to which subsidies and penalties affect product traceability, we increased the values of $F_l$ and $S_h$, respectively, to allow validity analyses under the parameterized conditions in Scenario 1. Figure 13 shows their evolutionary trajectories. It can be seen that, in the initial state, the PMs eventually choose the "not traceability" strategy ($F_l = 8, S_h = 10$). However, as the value of subsidies increases the PMs will choose the traceability strategy. Meanwhile, when government subsidies and penalties are raised by the same value, respectively, the final trend of the PMs converges to 1 when subsidies are raised, so PMs favor subsidies. In other words, increasing the subsidy $S_h$ for the traceability of manufactured products is more efficient in motivating PMs to choose the "traceability" strategy than increasing the penalty $F_l$ by the same value.

Moreover, as shown in Figure 14, the PSs finally choose the "no traceability" strategy ($F_n = 12, S_p = 15$) in the initial state. When the value of the subsidies increases the PSs do not change their strategy and still do not choose the "traceability" strategy. However, when the value of penalties increases the PSs will choose the "traceability" strategy. Compared to increasing the same amount of subsidy, encouraging suppliers to choose the "traceability" strategy is more likely to be achieved by increasing the penalty. It is worth noting that the evolutionary trajectory of the PSs fluctuates as the penalties increase. Therefore, the LGs need to add more penalties in order for the evolutionary trajectory of the PSs to converge to 1.

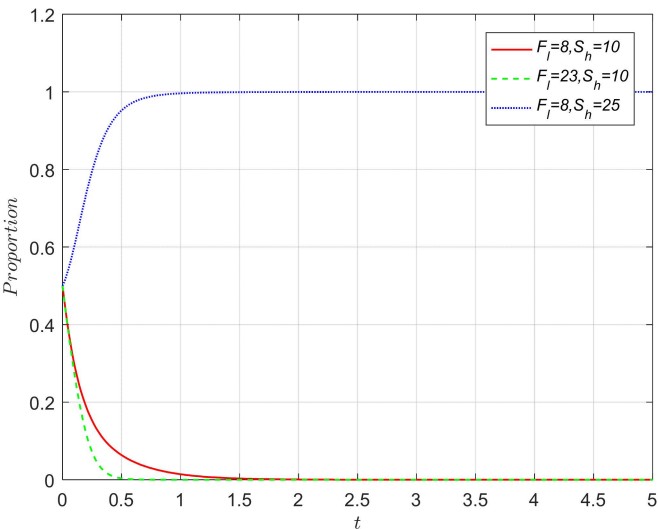

**Figure 13.** The effectiveness of subsidies and penalties for manufacturers.

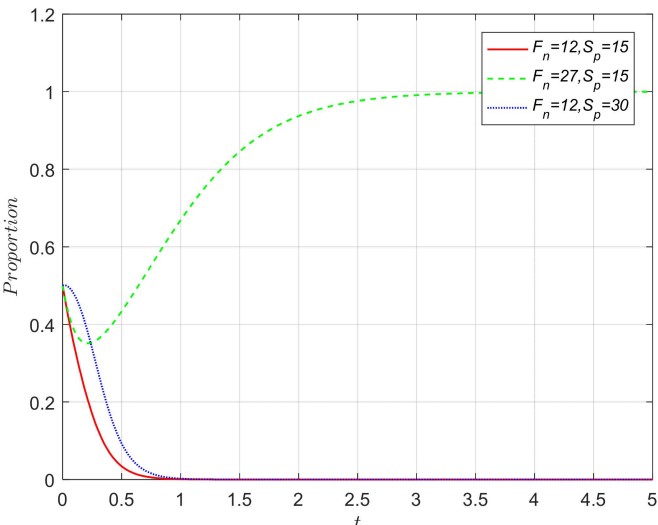

**Figure 14.** The effectiveness of subsidies and penalties for suppliers.

## 6. Conclusions and Policy Implications

### 6.1. Conclusions

The blockchain enables acquisition of the complete carbon footprint of the entire supply chain, creating traceable carbon emissions within it [11]. This capacity to trace carbon can help detect production links with high-carbon emissions and establish low-carbon supply chains, thereby bolstering public confidence in low-carbon products. Moreover, the implementation of blockchain traceability systems necessitates participation from all stakeholders due to their distributed computing nature. In this study, a tripartite EGT model is established among the PMs, the PSs, and the LGs. The study utilizes equilibrium stability and numerical simulation analysis to explore the long-term process of evolution and the mechanisms of strategic adaptation of PMs, PSs, and LGs. Furthermore, the influence of changes in key factors such as free-riding returns on the strategic decision making of players is examined. This study offers an evolutionary mechanism for examining the relationship between blockchain technology adoption and carbon traceability strategies, expanding our comprehension of this connection. The paper presents valuable guidance for improving the carbon trading market and developing a low-carbon supply chain by utilizing blockchain technology. Furthermore, for companies in the supply chain that participate in CF traceability, this study suggests the following implications.

The impact of penalties on the "traceability" behavior of suppliers is actually stronger compared to subsidies, contrary to common sense. However, subsidies have a more significant effect on manufacturers' "traceability" behavior when compared to penalties.

The manufacturer's sensitivity to the benefits and costs of the traceability strategy is evidently higher than that of the supplier. Changes in the benefits and costs of a manufacturer's traceability strategy have a greater impact on the probability of adopting the strategy than suppliers'. Hitch-hiking efficiency is a critical factor for both parties in deciding whether to adopt the "traceability" strategy. If the efficiency of "free-riding" exceeds the threshold value then the adoption of traceability will eventually evolve into a stabilizing strategy, which will converge to zero. It is apparent that a market with a strong low-carbon preference encourages manufacturers to adopt the "traceability" strategy.

Through the equilibrium stability analysis, combined with the actual situation, the ideal ESS is "the PMs choose traceability strategies, the PSs choose traceability strategies, and the LGs choose the strict regulation strategy". Local governments can avoid free-riding through incentives and penalties, as well as provide blockchain technology support, and can encourage supply chain stakeholders to choose traceability strategies [59]. After analyzing the operating conditions of the EGT system, it was discovered that the system can converge to the ideal state under certain conditions. The evolutionary equilibrium point is (1, 1, 1).

### 6.2. Policy Enlightenments

The paper offers the following policy insights based on these findings:

For the government, in order to improve supply chain transparency surrounding carbon emissions and improve environmental regulatory policies, the government should support the adoption of the blockchain traceability system through subsidies and penalties. The government should increase penalties for suppliers who fail to adopt a traceability strategy and secondarily consider increasing subsidies. The implementation of policies is crucial to enhance support for digital technologies and traceability platforms, thereby reducing the blockchain technology adoption cost. For the manufacturer, the government should develop policies to shape markets with low-carbon preferences and provide funding to install carbon capture and other infrastructure in high-carbon industrial clusters to track indirect emissions and ensure the integrity of carbon traceability. The government should provide consulting services to deliver blockchain solutions through blockchain technology incubation centers [60]. The above initiatives will encourage stakeholders in the low-carbon supply chain to actively participate in the blockchain traceability of carbon emissions, improving the transparency and traceability of their carbon footprints.

For manufacturers, first, they should set carbon emission limits and include carbon emissions, including energy consumption, in carbon footprint management. Second, through blockchain traceability systems, they should identify key production links with high-carbon emissions and improve processes or use environmentally friendly raw materials to reduce the amount of carbon emitted per unit of goods to reduce carbon tax costs. Finally, they should build a carbon traceability brand image to attract consumers.

For the suppliers, it is necessary to integrate all production links in the supply chain into the traceability platform and ensure that all carbon emissions data, including Scope 3, can be smoothly stored in the blockchain traceability platform. Additionally, suppliers should share the cost of infrastructure deployment by increasing subsidies to upstream manufacturers.

In this paper, participants' strategy choices are mainly influenced by the expected benefit–cost equilibrium, retroactive subsidies, and the free-riding effect. Yet, the credibility of each participant and the effectiveness of government investment support for the technology are also important variables affecting strategy choice, which need to be further explored in follow-up research. In addition, there are still some barriers to low-carbon supply chains: there are many actors involved and the process is complex.

This paper examines only the relationship between behavior and decision making, between PMs, PSs, and LGs, while blockchain traceability implementation is applicable

to most supply chains with carbon emissions. Therefore, commodity supply chains that combine more practical scenarios may be the future research direction. In addition, low-carbon supply chains should need to take into account the economic value they generate. For this reason, identifying revenue and carbon emissions as dual objectives via modeling and validation may be an important research direction.

**Author Contributions:** Methodology, Y.Z.; Writing—original draft, C.Z.; Writing—review and editing, Y.X. All authors have read and agreed to the published version of the manuscript.

**Funding:** This research was funded by Heilongjiang Provincial Natural Science Foundation of China, grant number grant number LH2021F035.

**Institutional Review Board Statement:** Not applicable.

**Informed Consent Statement:** Not applicable.

**Data Availability Statement:** Data are contained within the article.

**Conflicts of Interest:** The authors declare that they have no known competing financial interests or personal relationships that could have appeared to influence the work reported in this paper.

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
