# Peer review of "Blockchain Traceability Adoption in Low-Carbon Supply Chains: An Evolutionary Game Analysis"

_sustainability, doi:10.3390/su16051817_

Round 1

Reviewer 1 Report

Comments and Suggestions for Authors

 Journal: Sustainability

Manuscript ID: Sustainability-2802527

Title: Block Chain Traceability Adoption in Low-carbon Supply Chains: An Evolutionary Game Analysis

This paper is a well-crafted and engaging piece, although some errors in grammar and spelling have been detected. Considering this, I recommend thoroughly proofreading to ensure the manuscript lacks language-related shortcomings.

Abstract

The abstract mentions mathematical modelling and simulation methods but lacks details on the specific models or simulations employed. It is advised for the authors to provide a brief overview of the methodology. Moreover, the abstract could benefit from more specific details on the findings. For example, what are the identified effective blockchain-based traceability strategies for low-carbon supply chain members? Providing a glimpse of key findings would generate more interest.

Introduction

There are a lot of times when the authors forget to add full stop after the reference. Moreover, this section is well written but it’s too long. The authors have provided conceptual framework introduction which should have been provided in the literature review. I would suggest the authors to rephrase the contributions. Furthermore, more references should be provided to support important statements. For example:

Copied from Manuscript: “The key to reducing carbon emissions is for manufacturers and suppliers to invest in improving all aspects of their supply chains, such as choosing low-carbon raw materials and reducing utility use, including investing in digital technologies.

In this context, carbon footprint (CF) traceability helps to increase the transparency of carbon emissions in the supply chain and the trust of external regulators.”

“The Blockchain Traceability Platform (Taas), launched by ANT GROUP, has realized food supply chain traceability by providing traceability services for tea and cold chain foods.”

“Some stakeholders may have concerns, such as software integration costs and staff training costs. It is worth noting that from the perspective of decision-making behavior of stakeholders in the supply chain, despite the importance of blockchain technology for establishing carbon traceability solutions, the research on the strategies of relevant agents to adopt blockchain technology remain limited.”

Through literature review and analysis, existing studies mainly focus on the characteristics of blockchain technology, Carbon performance of sustainable economies, advantages and challenges of blockchain traceability, and pay little attention to the gaming behavior of participants in the process of deploying blockchain technology-based carbon footprint traceability in the supply chain.”

Comment: Especially this statement, where the authors need to provide evidence of prior studies to support these points. Even the research gaps that are discussed in later paragraph is not supported with references.

Literature Review and Theoretical Framework

When conducting scientific research, it is advised for the authors to refrain from making any evaluative statements. Regrettably, this section contains numerous evaluative statements, a few of which are ideological. Please add refereces to support the literature presented in this section.

It is advised that authors add a paragraph explaining table 1.

2.4 Evolutionary Game Theory: In this sub-section, the authors explain this theory but an analysis of the relationship between EGT theory and the issue investigated in this work is missing. Neither in the theoretical framework nor in the interpretation of the results is this theory mentioned. Moreover, the authors failed to support why this theory is most suited for the current study. It is noted that the authors' work needs to include the presentation of a hypothesis, which is a crucial aspect of a research study.

Materials and Methods

It is recommended that the authors consider incorporating a clear hypothesis statement in their future research work to ensure a well-structured and comprehensive study. The hypothesis seems to have been tested, but no hypothesis is mentioned.

Numerical Simulation

There is no comment.

Conclusion

There is no comment.

Good luck.

Comments on the Quality of English Language

Overall, the quality of English is fine. The ideas and statements are effectively communicated. However, it is crucial to strengthen your arguments and lend credibility to key statements by incorporating relevant references. Including supporting literature will enhance the robustness of your content and provide readers with additional resources for further exploration. Please consider integrating appropriate references to underpin your important points throughout the document.

Author Response

Dear reviewers
  Thank you for reviewing my manuscript and giving your valuable suggestions. I have further improved the article based on your suggestions.

Reviewer 2 Report

Comments and Suggestions for Authors The main question of the study is the analysis of interaction and mutual influence between three entities (product manufacturers, product suppliers and local governments) in the decision-making process of the blockchain adoption for traceability in the manufacturing industry.   Original and relevant for research area are both the problem formulation of the influence of supply chain participants and local regulators on the blockchain adoption decision, and its solution based on the Evolutionary game theory. A research gap is exploring the role of supply chain participants and local regulators in the decision of the blockchain adoption for reducing carbon emissions.   Modeling the behavior of product manufacturers and product suppliers under various regulatory influences, taking into account the strategies of other players.   It is necessary to explain in more detail the advantages of the Evolutionary game theory for solving the research problem. It is also necessary to clearly formulate the research hypothesis.   The findings are consistent with the presented research results. The main research questions posed were resolved using specific experiments presented in section 4 of the article.   Most of the references are appropriate, but two references (16 and 42) are incomplete.    Table 1 is not mentioned or described in the text.  

Author Response

(The authors gave the same response as above.)

Reviewer 3 Report

Comments and Suggestions for Authors

The paper has used mathematical modelling and simulation methods for studying the decision-making process of product manufacturers (PM), product suppliers (PS), and local governments (LGs), considering the adoption of blockchain traceability in the manufacturing industry. In this paper, a tripartite EGT model is established among the PMs, the PSs, and the LGs, its good. Interesting thing is this study has utilized equilibrium stability and numerical simulation analysis for exploring the long-term evolutionary process and strategic adaptation mechanisms of PMs, PSs, and LGs. One contribution is unique that is it offers an unique mechanism for examining the relationship between the adoption of blockchain technology and carbon traceability strategies, expanding the comprehension of this connection. This limited study has shown the behavioral decision-making relationship between PMs, PSs, and LGs, while the adoption and implementation of blockchain traceability is applicable to most supply chains with carbon emissions.

The paper needed to be formatted properly and English grammar have to be checked. 

Future works have to be elaborated in conclusions with a paragraph at-least. 

Author Response

(The authors gave the same response as above.)

Round 2

Reviewer 1 Report

Comments and Suggestions for Authors

The manuscript has been revised according to the author's suggestions and is now ready for publication.

Thank you

Reviewer 3 Report

Comments and Suggestions for Authors

Now necessary changes are done.